# The Method of Multi-Angle Remote Sensing Observation Based on Unmanned Aerial Vehicles and the Validation of BRDF

Hongtao Cao [1,2], Dongqin You [2,*], Dabin Ji [2] , Xingfa Gu [2,3], Jianguang Wen [2], Jianjun Wu [1], Yong Li [1], Yongqiang Cao [1], Tiejun Cui [1,4] and Hu Zhang [4]

1    Academy of Eco-Civilization Development for JING-JIN-JI Megalopolis, Tianjin Normal University, Tianjin 300387, China; caoht@tjnu.edu.cn (H.C.); tiejun_cui@163.com (T.C.)
2    State Key Laboratory of Remote Sensing Science, Aerospace Information Research Institute, Chinese Academy of Sciences, Beijing 100101, China; jidb@aircas.ac.cn (D.J.); guxf@radi.ac.cn (X.G.)
3    National Engineering Laboratory for Satellite Remote Sensing Applications, Aerospace Information Research Institute, Chinese Academy of Sciences, Beijing 100101, China
4    School of Geographic and Environmental Sciences, Tianjin Normal University, Tianjin 300387, China
*    Correspondence: youdq@aircas.ac.cn

**Abstract:** The measurement of bidirectional reflectivity for ground-based objects is a highly intricate task, with significant limitations in the capabilities of both ground-based and satellite-based observations from multiple viewpoints. In recent years, unmanned aerial vehicles (UAVs) have emerged as a novel remote sensing method, offering convenience and cost-effectiveness while enabling multi-view observations. This study devised a polygonal flight path along the hemisphere to achieve bidirectional reflectance distribution function (BRDF) measurements for large zenith angles and all azimuth angles. By employing photogrammetry's principle of aerial triangulation, accurate observation angles were restored, and the geometric structure of "sun-object-view" was constructed. Furthermore, three BRDF models (M_Walthall, RPV, RTLSR) were compared and evaluated at the UAV scale in terms of fitting quality, shape structure, and reflectance errors to assess their inversion performance. The results demonstrated that the RPV model exhibited superior inversion performance followed by M_Walthall; however, RTLST performed comparatively poorly. Notably, the M_Walthall model excelled in capturing smooth terrain object characteristics while RPV proved applicable to various types of rough terrain objects with multi-scale applicability for both UAVs and satellites. These methods and findings are crucial for an extensive exploration into the bidirectional reflectivity properties of ground-based objects, and provide an essential technical procedure for studying various ground-based objects' in-plane reflection properties.

**Keywords:** UAV; multi-angle remote sensing; BRF; M_Walthall; RPV; RTLSR

## 1. Introduction

Bidirectional reflectance properties of objects are essential for the inversion of quantitative information in remote sensing methods, encompassing both directional spectral information and spatial structural characteristics of objects. The bidirectional reflectance distribution function (BRDF) is commonly employed to describe the spatial distribution of bidirectional reflectance, and finds extensive applications in atmospheric correction [1], albedo retrieval [2], and vegetation surveying [3], among others. Satellite-based techniques such as MODIS [4], VIIRS [5], and POLDER [6] observe the Earth's surface from multiple angles through orbital displacements or mechanical oscillations of sensors. These methods typically operate at spatial scales ranging from a few hundred meters to several kilometers. Ground-based approaches have been utilized to investigate specific objects' bidirectional reflectivity properties [7]. Although aviation-based methods have undergone preliminary testing and validation, their widespread application has been limited due to inflexibility and high costs [8].

In recent years, unmanned aerial vehicles (UAVs) equipped with sensors have emerged as a novel remote sensing platform. These UAVs can acquire high-resolution remote sensing images at a reduced cost, and are capable of planning flight routes and sensor viewing angles according to specific requirements [9]. Notably, the integration of Global Positioning System (GPS) and Inertial Measurement Unit (IMU) in UAVs enables measurement of view zenith angle and azimuth angle, while photogrammetry technology accurately corrects the external orientation elements (3D coordinates X, Y, and Z and beam angles $\psi$, $\omega$, and $\kappa$) of sensors [10,11]. These parameters play a crucial role in recovering observation geometry. Consequently, studying bidirectional reflectance distribution function (BRDF) on ground objects based on UAV remote sensing becomes more convenient and feasible.

The advantages of high-resolution, multi-view observations for UAV remote sensing have garnered significant international attention from scholars. Firstly, the UAV is equipped with various sensors for multi-view observations, including spectrometers [12,13], 2D multi-spectral cameras [10], 2D hyperspectral cameras, [14] linear hyperspectral cameras [15], etc. Secondly, diverse flight paths are employed for multi-angle observations such as variable-altitude hovering [16], fixed-altitude hovering [12], and crisscrossing [17]. Moreover, extensive research and comparison have been conducted on the BRDF model and bidirectional reflectivity properties of ground objects based on UAV multi-view remote sensing [14,18]. Furthermore, BRDF inversion is utilized to optimize radiance differences between spectroscopic images captured from different angles in order to enhance the accuracy and quality of vegetation classification, leaf area index inversion, chlorophyll content inversion, and surface Albedo inversion [3,19]. However, UAV remote sensing as a novel multi-view observation scheme necessitates thorough research and evaluation. The flight paths that have been studied are complex, and the angle sampling interval is not uniform. The zenith angle and azimuth sampling of the flight path still require optimization to achieve higher BRDF inversion accuracy. Previous studies have verified the important value of UAVs in BRDF inversion, but the reliability of BRDF inversion results is not comprehensive. The quantitative evaluation of the deviation between different BRDF models and the actual directional reflectance, especially in the hot spot direction, hot spot reflectance, zenith reflectance, and BRDF shape, is an important basis for multi-angle remote sensing applications of UAVs. Considering the structural differences of ground objects, the applicability of the BRDF model to different ground objects has been neglected in the past research. This introduces uncertainties in further studies and parameter inversions in remote sensing.

This paper focuses on optimizing zenith angle and azimuth sampling to simplify the multi-angle observation flight paths, as well as to validate the performance of different BRDF models inversed from more aspects to clarify the applicability of different structural features of BRDF models. Section 2 presents the essential components and computational methods for BRDF inversion based on multi-view observations from UAVs. Section 3 describes the experiments conducted for multi-view observations, including details about the UAVs and sensors used, flight path design, and radiometric correction strategy. Section 4 discusses the experimental results and inverted BRDF using the M-Walthall model, RPV model, and RTLSR-model for various objects. The accuracy and structural properties of the inverted BRDF are evaluated using the measured data. Finally, Sections 5 and 6 provide a comprehensive discussion of the study's findings, followed by concluding remarks.

## 2. Methods

According to the definition of BRDF, it represents a mathematical function that characterizes the variation in reflectivity as a function of both incident and observation directions [20]. Consequently, the inversion of BRDF necessitates addressing three fundamental aspects: observed geometry, incident geometry, and bidirectional reflectivity. The acquisition of these three aspects can be achieved by UAV, as depicted in Figure 1, enabling the measurement of bidirectional reflection characteristics of ground objects.

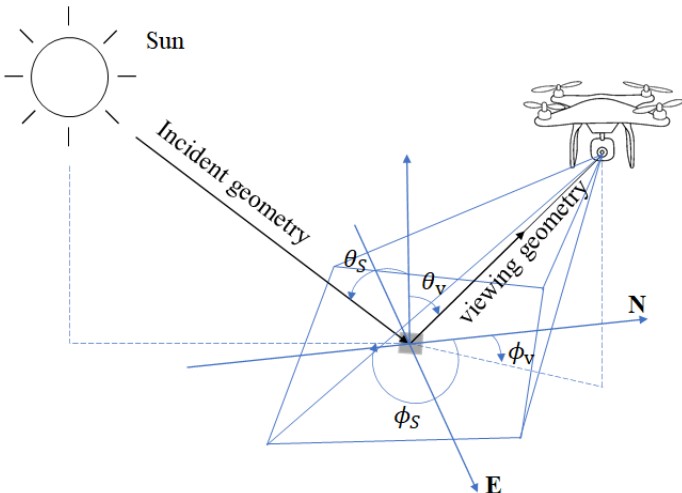

**Figure 1.** Schematic diagram of bidirectional reflection elements.

### 2.1. Observational Geometry of the Camera

The positioning and beam angle of the sensors are essential for establishing the observation geometry. The conventional approach involves measuring the view angle, which poses challenges to device performance. However, with advancements in photogrammetry, there is a compelling solution to determine the position and beam angle of aerial remote sensing cameras mounted on UAVs for capturing 2D spectral images. Typically, UAVs integrate cost-effective GPS and IMUs to establish the Position and Orientation System (POS), facilitating the simultaneous acquisition of approximate camera positions and orientations while capturing multi-view images [11]. By applying digital photogrammetry theory, accurate calculations can be made to determine both external and internal orientation elements of the camera. Please refer to Figure 2 for an illustration of this procedure.

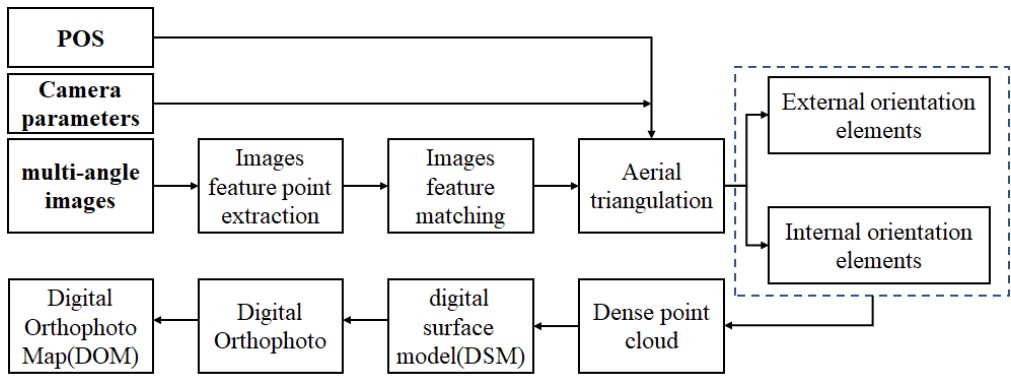

**Figure 2.** Flow chart of photogrammetry technology.

Based on the above, the spatial position of the camera and the target can be derived in the geodesic coordinate system. The zenith angle and azimuth angle between the camera and the target can be calculated using a geospatial location-based trigonometric function based on the collinearity equation consisting of the photographic center, image points and object points, as shown in Figure 3.

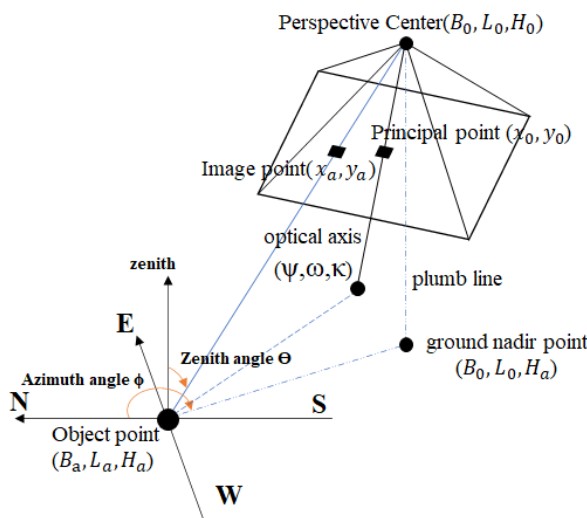

**Figure 3.** Zenith angle and azimuth angle of the observation beam of the camera.

Therefore, after knowing the camera's spatial position $(B_0, L_0, H_0)$ and target coordinate position $(B_a, L_a, H_a)$, the calculation formula for them are as follows:

$$\theta_v = \pi - \arctan\left(\frac{H_0 - H_a}{\sqrt{(B_0 - B_a)^2 + (L_0 - L_a)^2}}\right) \tag{1}$$

$$\phi_v = \begin{cases} \arctan\left(\frac{|L_0 - L_a|}{|B_0 - B_a|}\right) & L_0 - L_a > 0, B_0 - B_a > 0; \\[2mm] \pi - \arctan\left(\frac{|L_0 - L_a|}{|B_0 - B_a|}\right) & L_0 - L_a > 0, B_0 - B_a \leq 0 \\[2mm] \pi + \arctan\left(\frac{|L_0 - L_a|}{|B_0 - B_a|}\right) & L_0 - L_a \leq 0, B_0 - B_a \leq 0 \\[2mm] 2\pi - \arctan\left(\frac{|L_0 - L_a|}{|B_0 - B_a|}\right) & L_0 - L_a \leq 0, B_0 - B_a > 0 \end{cases} \tag{2}$$

$\theta_v$ is the zenith angle observed by the camera, and its values range from 0 to 90 degree; $\phi_v$ is azimuth angle observed by the camera, starting from north and rotates clockwise from 0 to 360 degrees.

### 2.2. Incident Geometry of the Sun

Due to the relative motion of the Earth and the Sun, the position of the Sun changes with the time of day and season, usually expressed as zenith Angle and azimuth in the horizon coordinate system as shown in Figure 4a. A shift in the position of the Sun not only causes a change in the intensity of the radiation reaching the surface, but also constitutes a different incident geometry. The position of the Sun can be measured by instruments or predicted by astronomical epochs.

Astronomers have been able to accurately predict the position of the Sun and Earth based on time, based on the laws of their orbits [21,22]. In astronomy, the equatorial coordinate system is commonly used to describe the position of the Sun, as shown in Figure 4b. The equatorial coordinate system extends the longitude and latitude coordinate system on Earth to the celestial sphere. A circle of latitude parallel to the equatorial plane is called a declination circle on the celestial sphere, and a circle of longitude passing between the north and south poles is called a time circle on the celestial sphere. In the equatorial coordinate system, the position of the Sun can be represented in terms of declination and time angle.

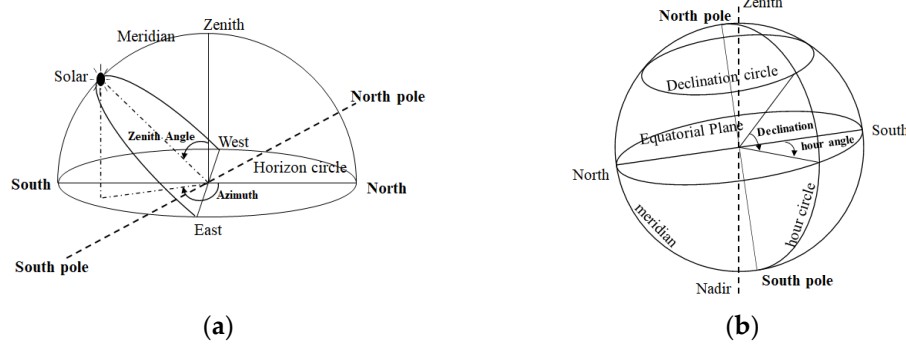

**Figure 4.** Position of the Sun: (**a**) solar altitude and azimuth in horizon coordinate system; (**b**) declination and time angle in equatorial coordinate system.

Meeus Jean provided an algorithm to calculate the declination and time angle of the sun, assuming that the earth's motion is a standard ellipsoidal orbit, ignoring the influence of the moon and other planets, and its accuracy is 0.01° [21].

In remote sensing observations, the zenith angle and azimuthal angle of the Sun in the horizon coordinate system are normally used to describe its position, which is based on the line-of-sight coordinates of the observation site. The position of the Sun is related to the observation date, time, and target spatial position. Based on the location of the target in the geographic coordinate system, the zenith angle and azimuth angle of the Sun at any time are calculated as follows.

$$\cos \theta_s = \sin B \sin \delta + \cos B \cos \delta \cos \Omega \tag{3}$$

$$\tan \phi_0 = \frac{\sin \Omega}{\cos \Omega \sin B - \tan \delta \cos B} \tag{4}$$

$$\phi_s = \pi + \phi_0 \tag{5}$$

$\theta_s$ is the zenith angle of the sun;

$\phi_0$ is the solar azimuth angle starting from the south, which is positive from south to west and negative from south to east;

$\phi_s$ is the solar azimuth angle starting from the north, which is 360° clockwise from east to west;

B is the local geographical latitude, with positive values in the northern hemisphere and negative values in the southern hemisphere;

δ is the solar declination of the day in the equatorial coordinate system;

Ω is the solar time angle in the equatorial coordinate system at that time.

### 2.3. BRF of the Objects

It is relatively difficult to measure the infield irradiance. The BRF is commonly measured, which is defined as the ratio of the reflected radiance of the target to the radiance of the Lambertian reference panel (LRP) under the same incident conditions [10,22]. An LRP with known spectral reflectivity should be taken to calculate the BRF. For a spectral image taken by sensors loaded on UAV, the DN (digital number) of pixels express the spectral radiance reflected by objects on the ground. The BRF of each pixel can be calculated as follows:

$$R_{i,j,\lambda}(\theta_v, \phi_v, \theta_s, \phi_s) = \frac{DN_{i,j,\lambda}(\theta_v, \phi_v, \theta_s, \phi_s)}{DN^t_{RCP,\lambda}} \times R_{RCP,\lambda} \tag{6}$$

$R_{i,j,\lambda}(\theta_v, \phi_v, \theta_s, \phi_s)$ is the bidirectional reflectance of the $i, j$ pixels of objects in the image of band $\lambda$;

$DN_{i,j,\lambda}(\theta_v, \phi_v, \theta_s, \phi_s)$ is the DN of pixels in the band $\lambda$ image;

$\overline{DN_{RCP,\lambda}^t}$ is the average value of DN of LRP in the band $\lambda$ image;

$R_{RCP,\lambda}$ is the spectral reflectance of the LRP in the band $\lambda$.

Considering the variation of the solar irradiance with time during the observation, the LRP needs to be measured before and after the flight. The LRP at any time in flight can then be obtained by linear interpolation to improve BRF reliability.

$$DN_{RCP}^t = \frac{DN_{RCP}^{tn} - DN_{RCP}^{t0}}{tn - t0} \times (t - t0) + DN_{RCP}^{t0} \tag{7}$$

$t0$ is the time for shooting the LRP before takeoff;

$tn$ is the time for shooting the LRP after landing;

$DN_{RCP}^{tn}$ and $DN_{RCP}^{t0}$ are the image DNs of the LRP measured before takeoff and after landing respectively.

It should be noted that the spectroscopic images need to be preprocessed before converting the DN of the pixels into the bidirectional reflectance of the object. First, the images were radiometrically calibrated to eliminate noise such as dark current and camera vignetting effects. The images were then additionally normalized according to exposure time and ISO.

## 3. Experiments

### 3.1. Multi-Angle Observing Route

The zenith angle and azimuth angle of views were specifically arranged to ensure uniform and comprehensive sampling of angles. The route planning is shown in the following figure, when the altitude from zenith to target is set to 200 m. They consist of zenith viewpoints (ZVP) and four Octagon routes with different radius and height. The vertices of the Octagon are the shooting points. The route design is shown in Figure 5.

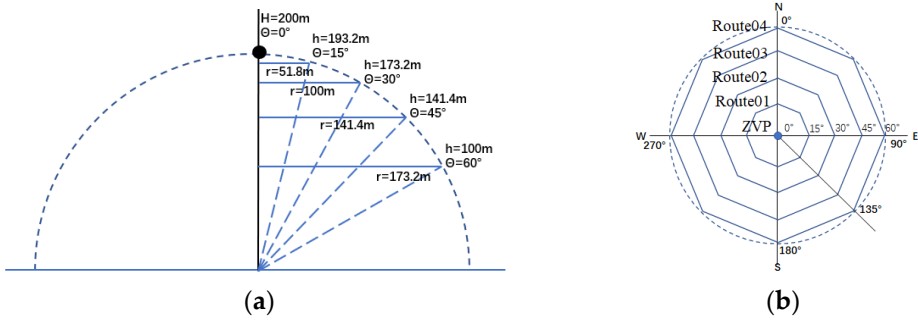

(**a**)  (**b**)

**Figure 5.** Design diagram for multi angle view: (**a**) sampling of zenith angle; (**b**) sampling of azimuth angle.

(1)    The flight route rotates with the target object as the center point to achieve sampling at different azimuth angles. Here, eight azimuth angles were planned with a sampling interval of 45° from 0 degrees to 360 degrees.

(2)    The radius and height of the rotating path determines the sampling of different zenith angles. Here, five zenith angles were designed with a sampling interval of 15° from 0 degrees to 60 degrees.

(3)    During this process, the optical axis of the camera always faces the target object, by adjusting the pitch angle of the sensor and the heading angle of the drone.

### 3.2. UAV Spectral Remote Sensing System

The DJI P4M remote sensing system was utilized to capture multi-angle spectral images. This system comprises a compact quadcopter drone equipped with a lightweight multispectral camera (Figure 6a). The multispectral camera is composed of six 2D CMOS sensors, including one RGB color sensor and five monochrome sensors that employ band-

pass filters to acquire five spectral images spanning the visible to near-infrared bands (Figure 6b) [23].

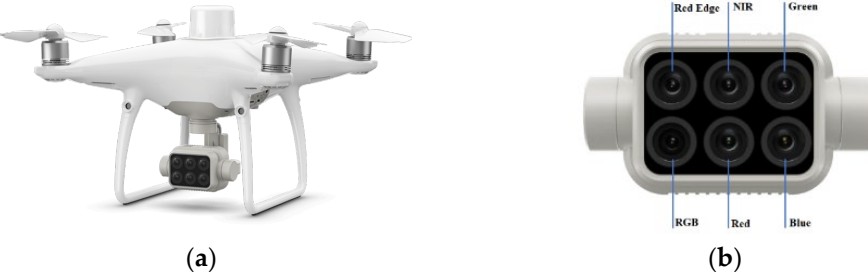

(**a**)                    (**b**)

**Figure 6.** Unmanned aerial vehicles used for multi angle reflectivity measurement: (**a**) DJI P4M remote sensing system; (**b**) multispectral camera loaded on DJI P4M.

After receiving the photographic instructions, five monochromatic CMOS sensors simultaneously capture images of the target in five distinct spectral bands. Their performance and characteristics remain consistent, with the exception of variations in their spectral response bands as outlined in Table 1.

**Table 1.** Performance and characteristics of five monochromatic CMOS.

| Parameters | Index |
|---|---|
| Controllable rotation range of PTZ | Pitch: −90° to +30° |
| Wave band of filters | Blue: 450 nm ± 16 nm; Green: 560 nm ± 16 nm; Red: 650 nm ± 16 nm; Red edge: 730 nm ± 16 nm; NIR: 840 nm ± 26 nm |
| FOV of lens | HFOV62.7° × VFOV50.9° IFOV 0.039° |
| focal length of lens | 5.74 mm (fixed) |
| Gain | 1×, 2×, 4×, 8× |
| Integral time | 1/100–1/10,000 s |
| shutter type | Global |
| Size of image | 1600 × 1300 (4:3.25) |
| Ground sampling distance (GSD) | 15.4 cm@ Relative Altitude = 200 m |

*3.3. Radiation Correction Programme*

The Lambertian reference panel (LRP) is characterized by its gray color and excellent stability properties, measuring 10 cm × 10 cm in size. The spectral reflectance of each band can be observed in Figure 7, provided below. Prior to UAV takeoff and after landing, the LRP should be examined twice to record the downward radiant intensity at both instances. Additionally, the camera will capture the time of image acquisition in the header file.

The observation area is equipped with two radiation reference panels (RRP) of varying reflectance to validate the reflectance images, each measuring 1 m × 1 m and exhibiting excellent stability and Lambertian properties. When capturing the target area, the spectral image will also include the RRP. The spectral reflectance for each band can be seen in Figure 8.

*3.4. BRF Reconstruction*

The experiments were conducted under clear and cloudless weather conditions. Following the predetermined flight routes, DJ P4M successfully acquired 33 sets of multi-angle spectral images. Additionally, two sets of LRP spectral images were captured on the ground before and after the flight.

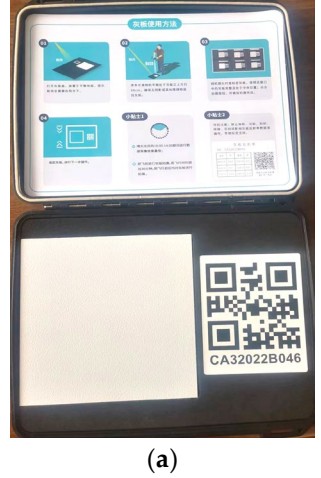
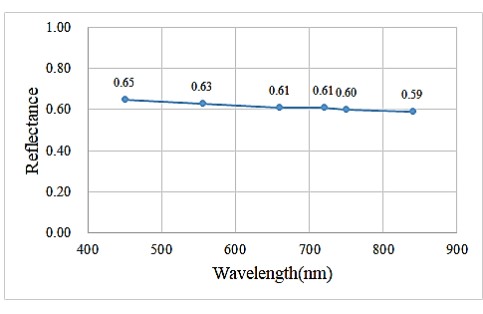

| (**a**) | (**b**) |

**Figure 7.** The Lambertian reference panel (LRP) for reflectance correction: (**a**) actual image of LRP; (**b**) spectral reflectance curve of LRP.

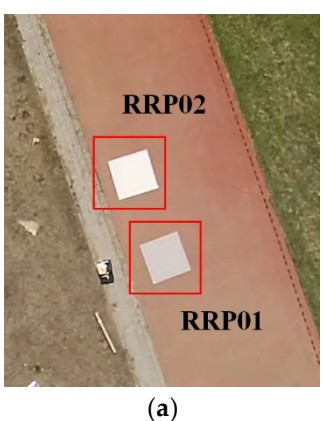
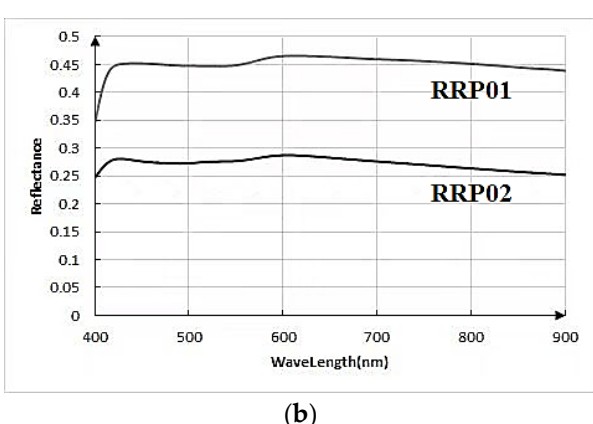

| (**a**) | (**b**) |

**Figure 8.** Radiation reference panels for reflectance correction: (**a**) actual images of RRP; (**b**) spectral reflectance curves of RRP.

### 3.4.1. Observational Geometry of the Camera

The multi-angle images were spatially corrected using Agisoft Metashape Professional software (Version 2.0.2 64 bit) (AMPS 2.0.2), which enables automatic image orientation and 3D modeling based on multi-angle photographs. The resulting external orientation elements of the camera, along with their corresponding accuracy, are presented in the table below. Figure 9 illustrates the consistent positioning and view angles of each corrected image.

The accuracy of the camera's external orientation elements was evaluated and is presented in Table 2. The relative accuracy of plane coordinates (X, Y) is at the centimeter level, while the relative accuracy of elevation (Z) is at the decimeter level. In comparison to GSD and IFOV, the external orientation elements remain within a distance equivalent to one pixel.

**Table 2.** Accuracy of the camera's external orientation elements calculated by AMPS 2.0.2.

| Accuracy Factors | X (m) | Y (m) | Z (m) | Omega (Degree) | Phi (Degree) | Kappa (Degree) |
|---|---|---|---|---|---|---|
| Mean Error | 0.063 | 0.063 | 0.118 | 0.040 | 0.036 | 0.021 |
| RMSE | 0.008 | 0.008 | 0.003 | 0.001 | 0.002 | 0.006 |

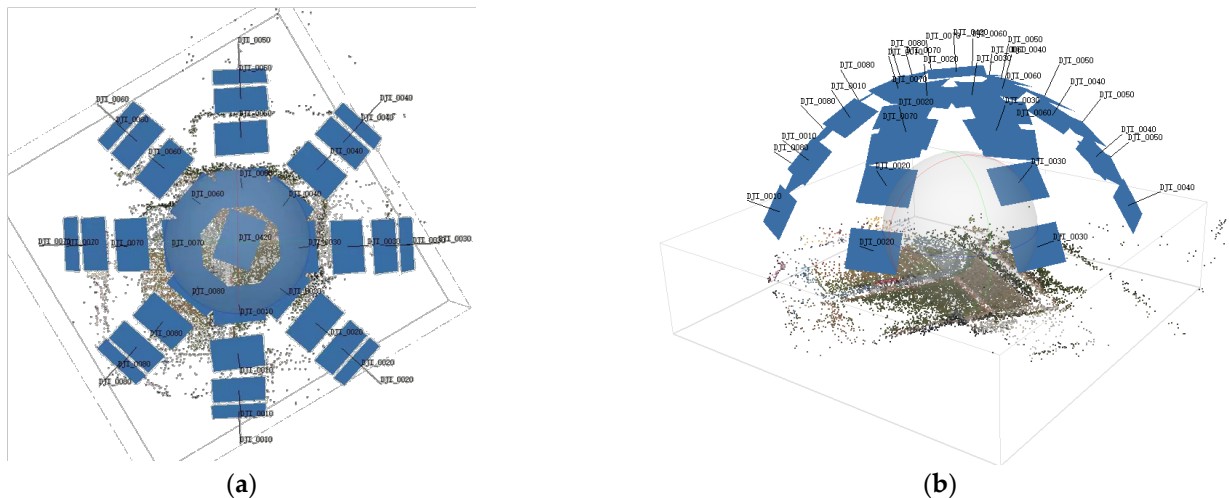

(**a**)                                                    (**b**)

**Figure 9.** Position and view-angle of each image corrected. (**a**) The top view displays the position and viewing angle of each image. (**b**) The side view displays the position and viewing angle of each image.

### 3.4.2. Incident Direction of the Sun

The duration of each route was brief, not exceeding 3 min. For an individual route, it is assumed that the variation in sun position has a negligible impact on reflectance directionality, and the midpoint time can be considered as the reference point. As per the methodology described in Section 2.2, Table 3 presents the sun positions for ZVP and each route.

**Table 3.** Time of ZVP and routes and corresponding solar zenith angle and azimuth.

| Routes of Flight | Duration | Middle Time | Zenith Angle of Sun (Degree) | Azimuth Angle of Sun (Degree) |
|---|---|---|---|---|
| ZVP | 10:08:40 | 10:08:40 | 39.46 | 112.00 |
| Route01 | 10:09:19–10:10:23 | 10:09:46 | 39.36 | 112.31 |
| Route02 | 10:11:29–10:12:54 | 10:12:04 | 39.15 | 112.97 |
| Route03 | 10:13:48–10:15:31 | 10:14:31 | 38.93 | 113.69 |
| Route04 | 10:16:35–10:18:31 | 10:17:23 | 38.69 | 114.54 |

### 3.4.3. Reconstructing the Geometric Structure of "Sun-Object-View"

After performing radiometric calibration and correction, the digital numbers (DNs) of 33 sets of multi-angle spectral images were converted into bidirectional reflectance factors (BRF). From the region with overlapping multi-angle images, four objects (RRP01, treetop, lawn, soil) were selected to construct their respective BRFs. RRP01 is characterized by a very flat and smooth surface, while treetop, lawn, and soil exhibit flatness combined with roughness. The plane position and elevation of these objects were extracted from the digital orthophoto model (DOM) and digital surface model (DSM), as described in Section 2.1. This information was then used to calculate the view zenith angle and azimuth angle based on Formulas (1) and (2). The positions of these objects on the true color DOM are illustrated in Figure 10.

The bidirectional reflectance factor (BRF) of each pixel in the spectral images was computed as described in Section 2.3. A window of pixels with a size of 3 × 3 was extracted and averaged to obtain the BRF for the selected objects. The BRF values for these objects were determined based on the camera's observational geometry, considering specific zenith and azimuth angles. Taking the NIR band (840 nm) as an example, Figure 11 illustrates the spatial distribution of reflectance for these objects.

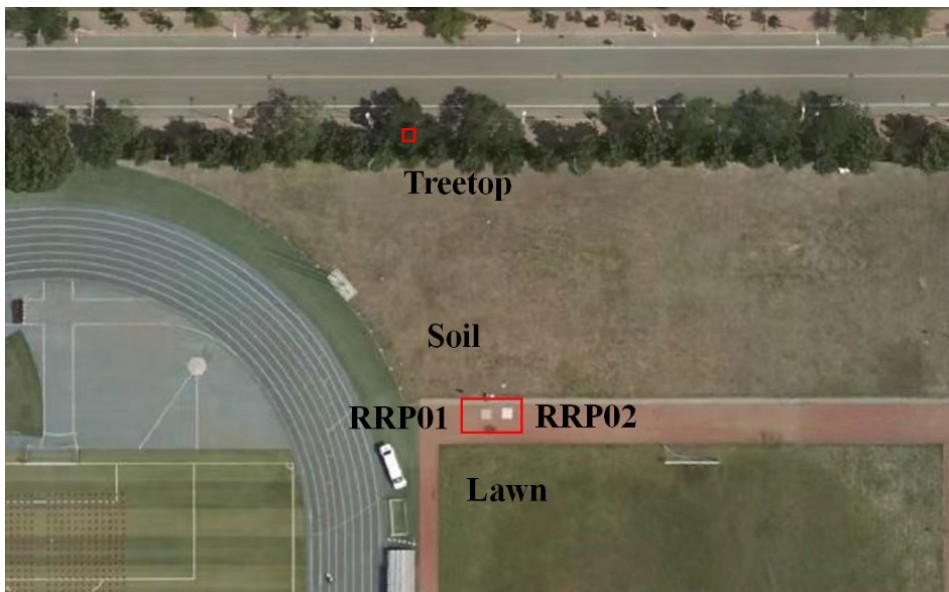

**Figure 10.** Four objects selected arrange in DOM.

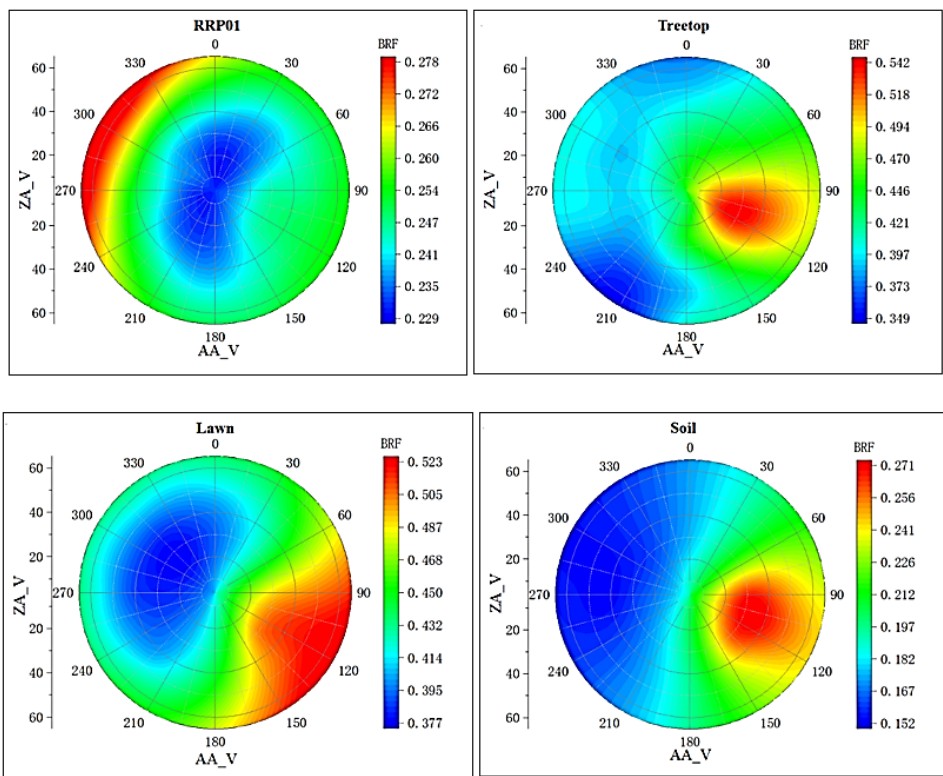

**Figure 11.** Spatial distribution of reflectance of four objects (ZA_V is the zenith angle of view, which ranges from 0 to 60 degrees; AA_V is the azimuth angle of view, which ranges from 0 to 360 degrees).

The structure exhibits the following characteristics:

(1) For the smooth RRP01, its bidirectional reflectance factor (BRF) assumes a bowl-shaped form with stronger forward scattering than backscattering.

(2) In contrast, for the rough treetop, lawn, and soil surfaces, their BRFs display greater complexity with stronger backscattering compared to forward scattering.

(3) Furthermore, all of these BRFs demonstrate nearly symmetrical behavior along the principal plane of the sun.

## 4. Inverting and Validating BRDF

UAV remote sensing has facilitated the study of bidirectional reflectance at a scale ranging from centimeters to decimeters. In order to determine its feasibility for UAV remote sensing and the reliability of multi-angle routes, classic BRDF models used in satellite-based and ground-based remote sensing were inverted and validated. Currently, widely used BRDF models include M-Walthall, RPV (Rahman Pinty Verstraete), and RTLSR (RossThick-LiSparseR).

(1)　Walthall is an empirical model proposed by Walthall et al. (1985) based on extensive field experimental data for correcting soil BRDF. Nilson and Kuusk (1989) modified it to adhere to the reciprocal principle, resulting in M-Walthall [17,19].

(2)　RPV is a semi-empirical model that incorporates the Henyey Greenstein scattering phase function and hot spot effect term to ensure consistency with actual bidirectional reflection using the Minnaert empirical model [24–27].

(3)　RTLSR is a nuclear-driven model formed by combining Ross Thick core and LiSparseR core, where the former serves as the volume scattering core in this nuclear-driven model, while the latter acts as the geometrical optics core. It has been widely employed in producing satellite remote sensing BRDF/Albedo products [5,28,29].

The three BRDF models each involve 4 and 3 unknown coefficients, respectively. Consequently, the coefficients of these models can be determined by employing redundant observation data through the least square fitting.

RRP01 along with three objects were selected as experimental targets, which aligns with Figure 10 of this paper. The least square method was utilized to fit each target's 33 sets of BRF data into these three BRDF models. These 33 sets of data for each target are reflectance samples in 33 directions, indicating the spatial variation of the target's bidirectional reflection.

### 4.1. Accuracy of BRDF Fitted

The correlation coefficient, obtained through the least square method, serves as an indicator of the accuracy of BRDF inversion. A value closer to 1 signifies a higher quality fitting. As presented in Table 4, all three BRDF models exhibit commendable fitting accuracy. RPV demonstrates the highest level of precision, followed by M-Walthall, while RTLSR exhibits comparatively lower accuracy.

**Table 4.** Correlation coefficient fitted by least square method for three BRDF models.

| BRDF Model | RRP01 | Treetop | Lawn | Soil |
|---|---|---|---|---|
| M-Walthall | 0.794 | 0.809 | 0.874 | 0.850 |
| RPV | 0.825 | 0.901 | 0.959 | 0.925 |
| RTLSR | 0.647 | 0.257 | 0.848 | 0.621 |

The consistency between the simulated reflectance using BRDF models and the measured reflectance is illustrated in Figures 12–15. The obtained results align with the aforementioned description.

### 4.2. Structure of BRDF

BRDF models can be developed to calculate BRF for a range of zenith angles from 0 to 60° and azimuth angles from 0 to 360°, based on the given sun's zenith angle of 39.15° and azimuth angle of 112.97°. These BRDF coefficients can then be used to construct polar coordinate representations of the BRDF models, as illustrated in Figures 16–19.

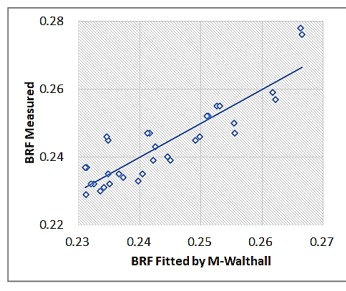 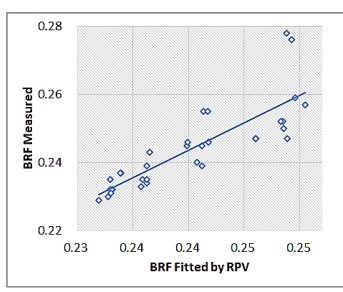 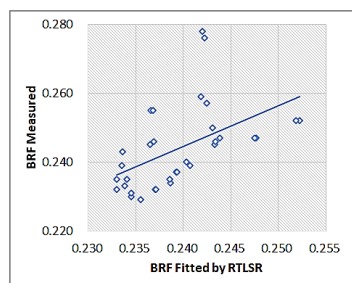

**Figure 12.** Correlation of RRP01 inverted with M-Walthall, RPV, RTLSR.

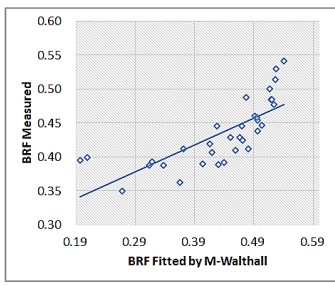 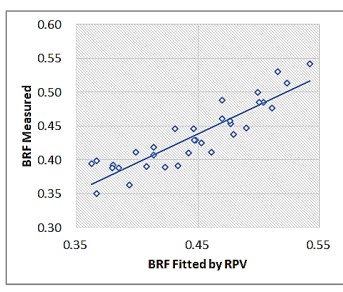 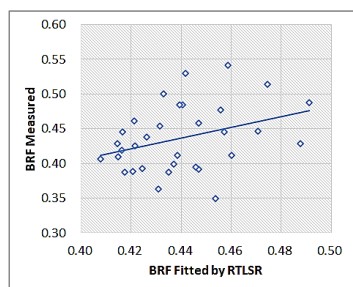

**Figure 13.** Correlation of RRP02 inverted with M-Walthall, RPV, RTLSR.

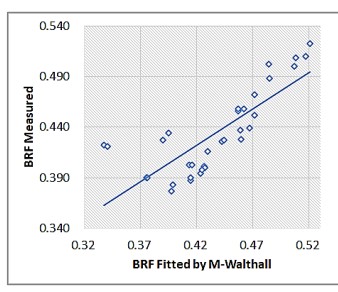 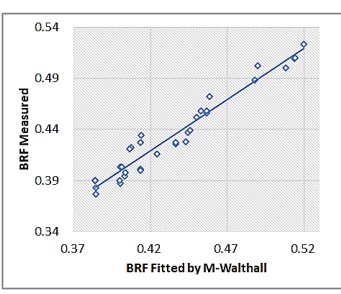 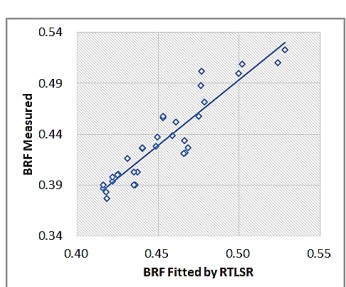

**Figure 14.** Correlation of lawn inverted with M-Walthall, RPV, RTLSR.

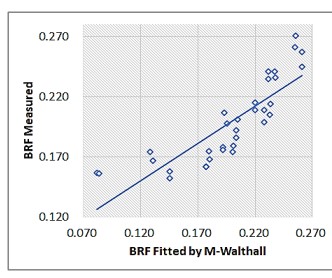 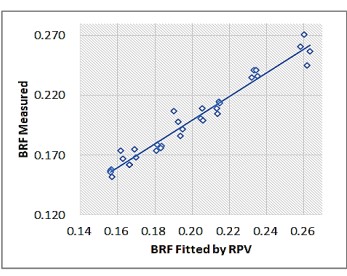 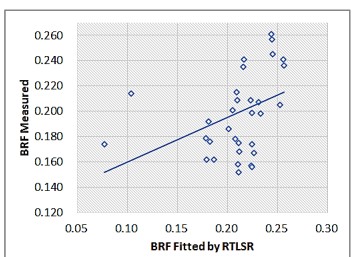

**Figure 15.** Correlation of soil inverted with M-Walthall, RPV, RTLSR.

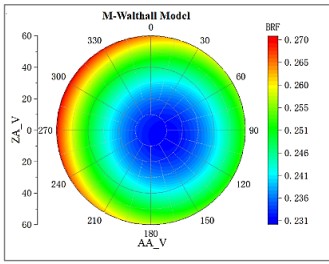 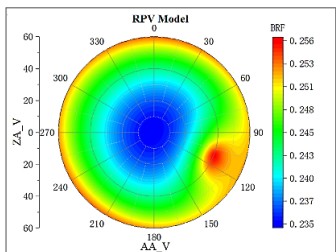 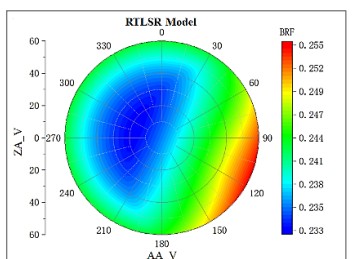

**Figure 16.** BRDF of RRP01 inverted with M-Walthall, RPV, RTLSR.

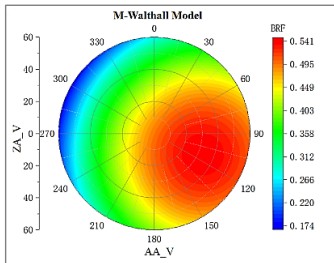 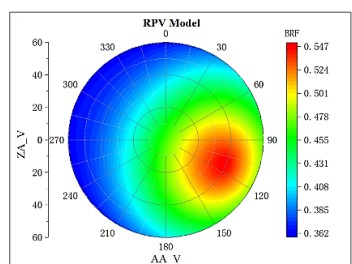 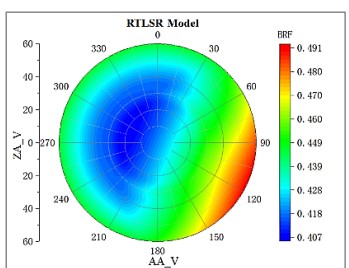

**Figure 17.** BRDF of treetop inverted with M-Walthall, RPV, RTLSR.

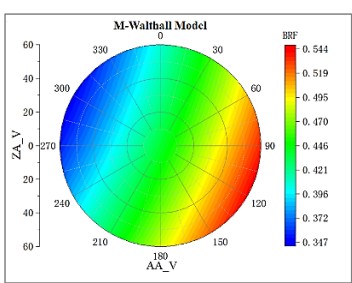 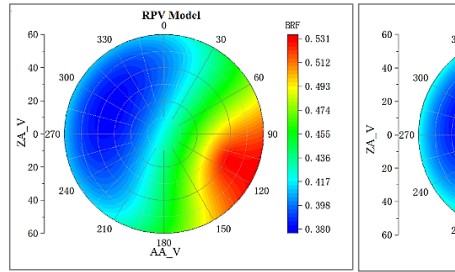 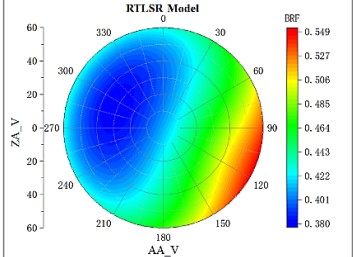

**Figure 18.** BRDF of lawn inverted with M-Walthall, RPV, RTLSR.

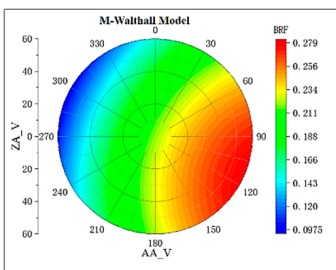 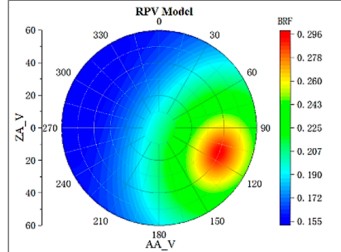 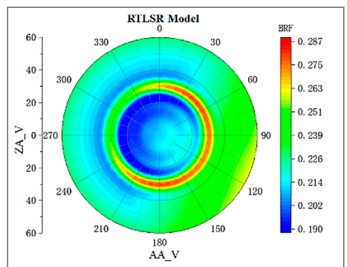

**Figure 19.** BRDF of soil inverted with M-Walthall, RPV, RTLSR.

Comparing with Figure 11, discrepancies can be observed in the fitted bidirectional reflectance distribution function (BRDF) for the same object. For smooth RRP01, M-Walthall's fitting yielded the closest resemblance to the measured data. Conversely, for rough objects, RPV's fittings exhibited a higher similarity to the measured BRDFs compared to those obtained by M-Walthall, which were only moderately similar. Notably, RTLSR's fitting lacked any significant similarity, and merely achieved a consistent structure of BRDF on the lawn surface. To further validate the structural characteristics of BRDF, profile values within 0–180 degrees and 90–270 degrees were extracted. Figure 20 illustrates the trend line depicting measured and fitted values near profiles within 0–180 degrees; while Figure 21 presents a similar trend line near profiles within 90–270 degrees.

The profile trend lines of BRDFs exhibit strong consistency between the values fitted by M_Walthall and RPV models and those measured, with the RPV model demonstrating superior performance.

### 4.3. Hotspot of BRDFs

An essential characteristic of directional reflectance models is the presence of a hotspot, which represents a peak in reflectance near the observation direction that aligns precisely with the incident direction of sunlight. Figure 22 illustrates the extracted and displayed BRF values corresponding to the sun's primary plane.

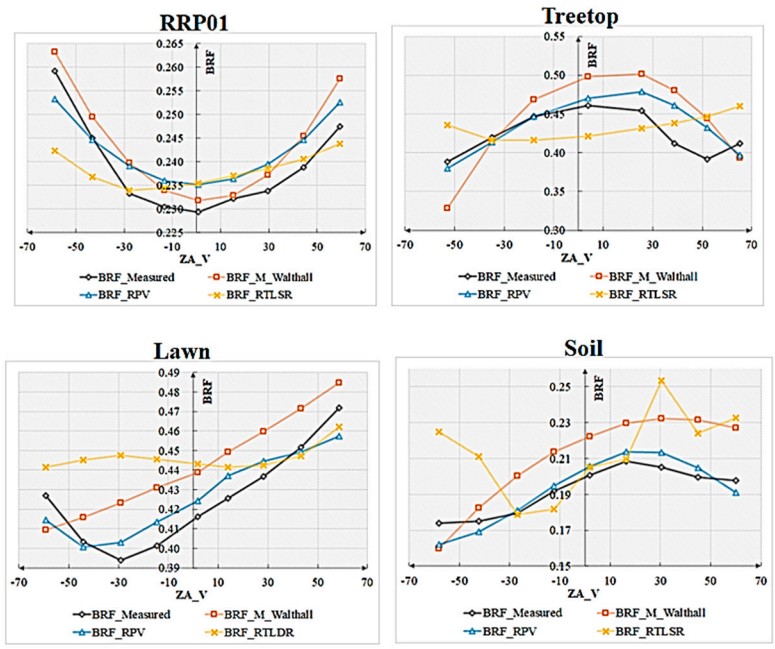

**Figure 20.** BRDF profiles at 0–180 degrees.

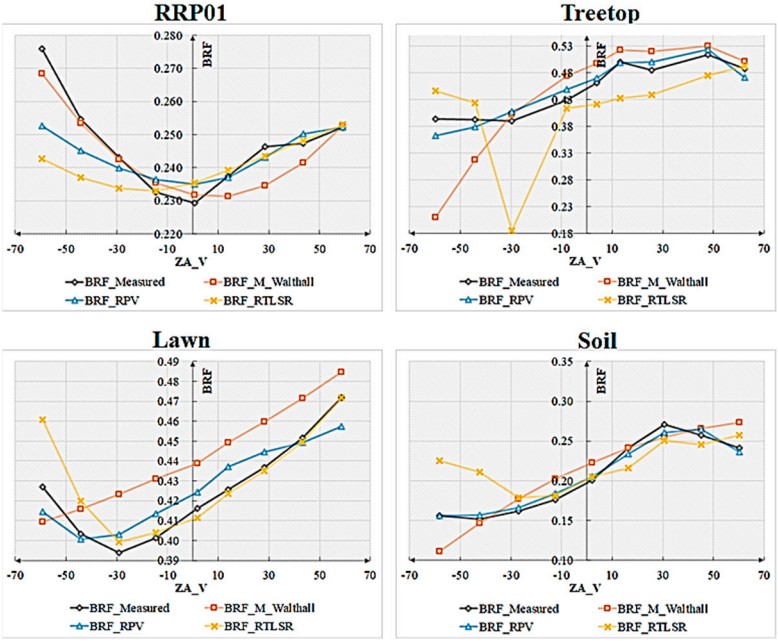

**Figure 21.** BRDF profiles at 90–270 degrees.

Among the three BRDF models, only the RPV model exhibits pronounced hotspots that align consistently with the solar direction. It is worth noting that while smooth RRP01 appears to display hotspots, this observation contradicts the measured BRF data.

### 4.4. Errors of Reflectance Values Fitted and Measured

Additionally, in order to assess the quantitative disparities between the fitted model and the measured data, a comparison was made between the mean reflectance (zenith angle 0–60 degrees, azimuth angle 0–360 degrees) and zenith reflectance values. The absolute errors of both the mean reflectance and zenith reflectance values were depicted in Figure 23. The error is defined as the discrepancy between the measured reflectance and the value calculated by the BRDF model. In this case, we are referring to the absolute error, since reflectivity is a dimensionless quantity and does not have any units.

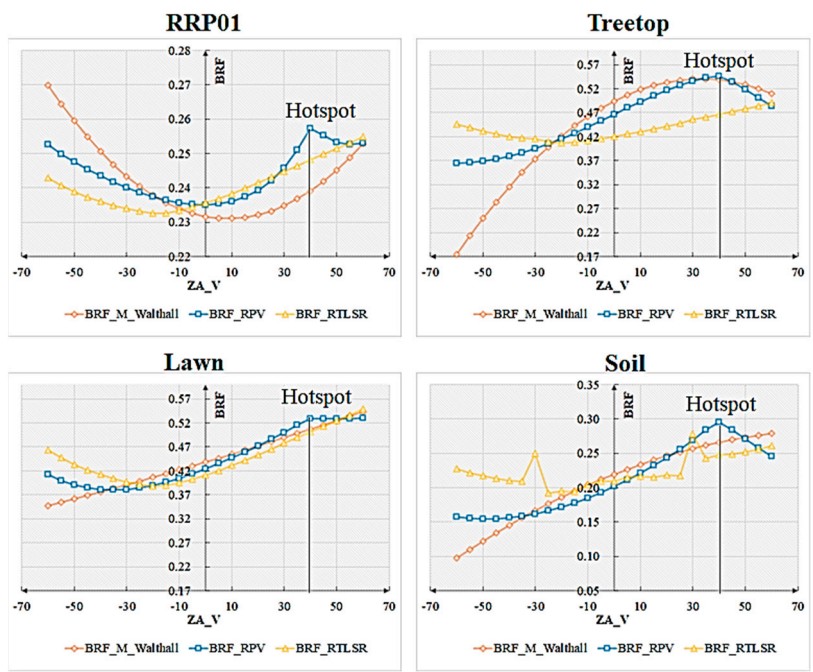

**Figure 22.** Hot spot effect in the solar principal plane reproduced by three BRDF models.

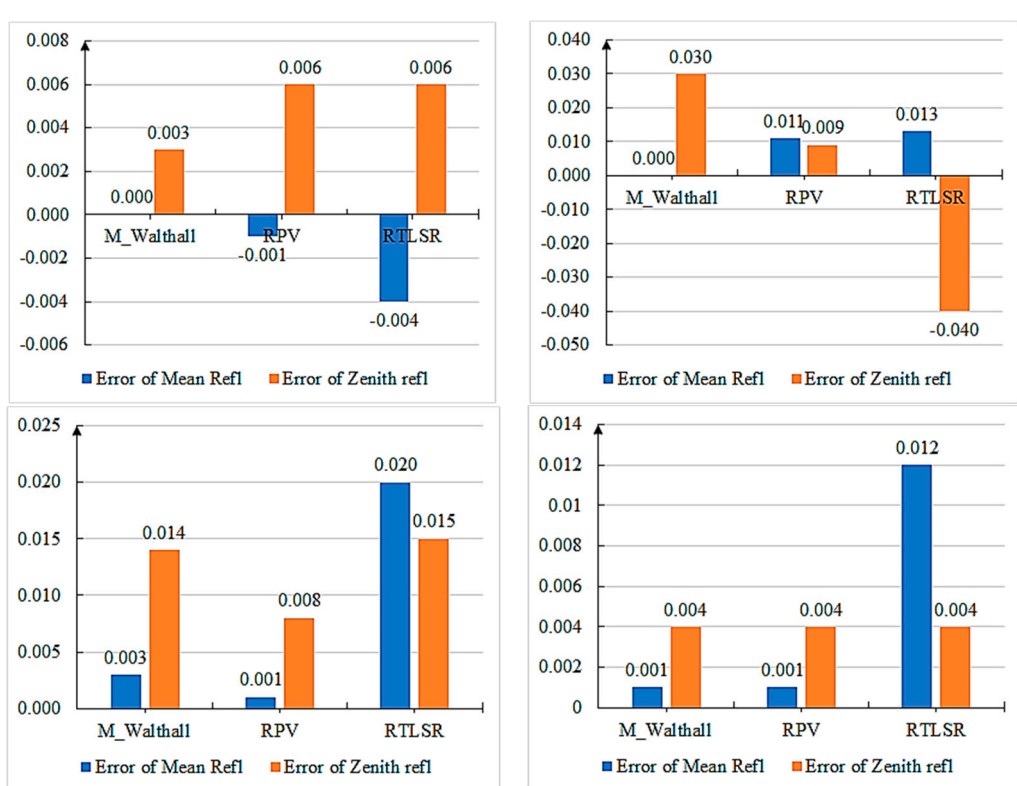

**Figure 23.** Error of mean reflectance and zenith reflectance.

Overall, the RPV model exhibits smaller errors in both mean reflectance and zenith reflectance on rough surface features. The M_Walthall model demonstrates minimum error on a smooth RRP01 surface. Conversely, the RSTLSR model generally yields high errors.

## 5. Discussion

Currently, UAV remote sensing is undoubtedly a newcomer in the field of remote sensing. Its characteristics of cost-effectiveness, high resolution, and flexibility have at-

tracted many scholars to pursue it, leading to its wide application in surface classification, water quality monitoring, and forest surveying [30,31]. However, the ability of traditional instruments and remote sensing satellites to observe from multiple angles is limited due to the complexity of changing positions and angles. Spectrometers for multi-angle measurements are excessively heavy and challenging to apply for extensive field measurements. Satellite-based multi-angle sensors like MISR, MODIS, and VIIRS require combining data from multiple orbits to achieve multi-angle measurement, but are inevitably influenced by surface changes and atmospheric variations [32]. Moreover, their spatial scale fails to meet the BRDF characteristics of pure ground objects. With advancements in unmanned aerial vehicle (UAV) technology and photogrammetry techniques, utilizing multi-angle remote sensing observations and BRDF inversion has become feasible. Nevertheless, in order to improve the potential of UAV multi-angle remote sensing, further improvements and validation are needed, such as optimizing flight plans for multi-angle observations or automating image matching processes while considering scale BRDF models.

In this study, a novel method of multi-angle observation using unmanned aerial vehicles (UAVs) is adopted and validated. A polygonal flight path is designed to cover the hemispherical zenith angle sampling and omnidirectional angle sampling. This design ensures a more balanced angular spacing of bidirectional reflectance distribution function (BRDF) samples, meaning they can maintain a consistent distance from each other. Compared to fixed-altitude hovering or parallel routes, this approach allows for observations at zenith angles greater than 60 degrees. The simplicity and effectiveness of this have been demonstrated. Additionally, the external azimuth elements obtained through aerial triangulation provide essential conditions for the geometric reconstruction of observation data. In this paper, we analyze the accuracy and determine the reliability of multi-angle image matching. Consequently, two-dimensional multi-angle remote sensing does not require precise angle measurements or recording, simplifying the measurement procedure. Furthermore, we fit classical BRDF models based on different scales (Malthall model, RPV model, and RTLST model), comparing their performance at the UAV scale. Our findings indicate that the RPV model exhibits superior performance in UAV remote sensing applications, and is suitable for ground objects with diverse structures. Finally, the RPV models are also applicable to satellite-scale BRDF inversion due to their multi-scale adaptability.

This study presents technical procedures and recommendations for selecting a BRDF model in remote sensing reflectivity studies, aiming to address the challenges and limitations associated with multi-angle observations from ground-based and satellite remote sensing platforms. The utilization of UAVs simplifies BRDF measurements, offering a convenient approach to investigate the scale characteristics and type variations of bidirectional reflectance across diverse ground objects. Consequently, we anticipate that research on foundation materials' bidirectional reflectance will experience significant growth. In the future, UAVs are expected to play an increasingly prominent role in studying the bidirectional reflection properties of ground objects, establishing databases for such objects, as well as validating satellite-based remote sensing radiation products. They will undoubtedly contribute to the advancement of remote sensing technology.

## 6. Conclusions

In this paper, we comprehensively study the flight routing, BRF reconstruction, and performance of BRDF models for multi-angle remote sensing with UAVs. With the help of photometric methods, the observed geometry of the sensor is reconstructed and the BRF pixels of the ground-based objects are spatially registered. The results indicate that polygonal flight routes designed along hemispherical space are effective and essential for studying the bidirectional reflectivity of ground objects. Three BRDF models, M_Walthall, RPV, and RTLSR, were selected to test their performance on the UAV scale. The three fitted BRDFs are compared with the measured BRF in terms of fitting quality, shape structure, and reflectivity errors. The results show that the RPV model has the best inversion performance, M_ Walthall comes second, and RTLST is the worst. The M_Walthall model performs well

on smooth terrain objects. RPV is applicable to various types of rough terrain objects and has multi-scale applicability for UAVs and satellites. These methods and conclusions are important for an in-depth study of the bidirectional reflectivity of ground-based objects.

**Author Contributions:** Conceptualization, H.C., D.Y., D.J. and X.G.; methodology, H.C., D.Y., X.G. and J.W. (Jianguang Wen); formal analysis, H.C., D.Y. and D.J.; writing—original draft preparation, H.C. and H.Z.; writing—review and editing, H.C., D.Y. and D.J.; supervision, Y.C., Y.L. and T.C.; project administration, J.W. (Jianjun Wu); funding acquisition, H.C. All authors have read and agreed to the published version of the manuscript.

**Funding:** This research was supported by the Open Fund of State Key Laboratory of Remote Sensing Science (Grant No. OFSLRSS202314).

**Conflicts of Interest:** The authors declare no conflict of interest.

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
