# Peer review of "The Method of Multi-Angle Remote Sensing Observation Based on Unmanned Aerial Vehicles and the Validation of BRDF"

_remotesensing, doi:10.3390/rs15205000_

Round 1

Reviewer 1 Report

This paper proposes a polygonal design of the UAV flight which is adequate to test the performance of three BRDF models : M_Walthall, RPV, RTLSR. Four types of surfaces are tested (radiation reference panels, soil, lawn and tree tops).

The contains a good presentation of the main BRDF concepts.

The English is good and clear. The paper is concise and well structured.

Minor corrections:

Figure 2: Define « POS » in the text.

Figure 3: “Plumb line” appears two times in the figure. I assume the right one is related to the “ground nadir point”. I don’t understand what is the second “plumb line” related to.

Figure 4: The “N” and “S” are represented in the same plane as the “Equatorial Plane”. The “Zenith” and “Nadir” aren’t, in this case the “N” and “S”?

Line 137: “Meeus Jean” need of citation.

Line 141: “transverse coordinate system” :  The projection you used is Universal transverse of Mercator?

Figure 11: What is the meaning of the plot labels “ZA_V” and “AA_V”? Specify also what represents the degrees and the units of the right axis.

Line 283: What is the meaning of “principal plane” here? Is it the solar principal plane?

Lines 302-304: I suggest you develop this paragraph with more details on how you apply these models.

Line 306: Please, explicitly specify the two variables that are used to generate the correlation.

Line 285: Number should be “4” instead of “3”.

Author Response

Dear expert,

Thank you very much for taking precious time to review my manuscript entitled “The Method of Multi-angle Remote Sensing Observation based on Unmanned Aerial Vehicles and the Validation of BRDF”.

I have revised the manuscript based on your comments and suggestions. Now, I am resubmitting the updated version along with annotations for further review.

Thank you and Best wishes.

Your sincerely,

First author: Hongtao Cao

Reviewer 2 Report

The Method of Multi-angle Remote Sensing Observation and the Validation of BRDF leverages the agility and flexibility of UAVs to collect multi-angle data, which is then used to validate and enhance BRDF models. This approach provides valuable insights into the reflectance properties of Earth's surfaces and has numerous applications in environmental and ecological studies. It is particularly useful for land cover classification and for understanding vegetation structure and health, which can be influenced by various factors like canopy orientation and moisture content.

The authors comprehensively studied the flight routing, BRF reconstruction and performance of BRDF models under clear and cloudless weather conditions. As not always we benefit of clear sky during surveys, do the authors have input for results in not-so-perfect weather?

Author Response

(The authors gave the same response as above.)

Reviewer 3 Report

Comments to Author(s)

The study is designed and well-explained in a way that will interest readers. However, the following corrections would make the study more understandable.

Line 16: unmanned aerial vehicles -> Unmanned Aerial Vehicles

Line 74: hots pot -> hot spot

Line 33: Literature review is weak. Studies in the literature should be compared more comprehensively and the originality of the publication should be demonstrated. 

Line 93: I did not see the citation for Figure 1.

Line 99: You have already explained UAV above. You should use an abbreviation.

Line 105: What is POS?

Line 315-335: Figures 12-15 and Figures 16-19 can be combined.

Line 369, 370: How did you find these errors and do they have units? If so, what are they?

Line 376: The Discussion section should be rewritten. In the Discussion section, the results obtained from the study should be evaluated in all aspects and a comparison with previous studies should be made (if necessary).

Author Response

(The authors gave the same response as above.)

Round 2

Reviewer 3 Report

Substantial changes have been made to the article after the first revision. Thank you for taking my comments into account and editing the article. I think the article would be more understandable with a few minor stylistic corrections. 

There is a clarity problem in almost all graphs. If you are getting your graphs from MS excel, a good solution would be to save them in .pdf format and then insert them into MS word.

Author Response

Dear expert,

Thank you very much for taking precious time to review my manuscript entitled “The Method of Multi-angle Remote Sensing Observation based on Unmanned Aerial Vehicles and the Validation of BRDF”.

I have revised the manuscript based on your suggestions. Now, I am resubmitting the updated version along with annotations for further review.

Thank you and Best wishes.

Your sincerely,

First author: Hongtao Cao
